# ContrastGen: A Multi-agent Contrastive Framework for Hard Retrieval Data Generation

## Abstract

The embedding model vectorizes queries and passages separately and uses the distance between the two resulting vectors as the basis for retrieval matching. It serves as a core component in retrieval tasks. However, since training datasets often consist predominantly of simple queries, the embedding model is usually unable to develop the capability to handle complex, hard queries. This leads to a serious performance bottleneck and an upper limit on its effectiveness. To address the challenge of handling hard queries, existing methods propose new training strategies tailored for embedding models or simplification mechanisms during the query inference phase. In contrast and orthogonal to these approaches, this paper tackles the problem from the data level, aiming to improve the performance of the embedding model by generating high-quality hard query training data. More specifically, inspired by the ability of agents to closely simulate human behavior, and with the goal of generating queries that retain semantics and logical knowledge similar to those of human-generated queries, this paper proposes a multi-agent framework to generate hard queries, thereby enhancing the training performance of the embedding model. The core idea involves first using a generation agent to create new queries, followed by specialized agents–such as those focused on logical reasoning and semantic understanding–to filter and identify truly hard queries. Experimental results on different embedding models and datasets demonstrate that our method outperforms existing approaches.

## 1 Introduction

The goal of an information retrieval task is to accurately find the matching passage from a large set of passages based on a given query. It has been widely applied across various domains. For example, in the e-commerce field (Zheng et al., 2023), systems match relevant products based on user queries, while literature search engines match appropriate documents according to user input (Liu et al., 2014). Currently, as shown in Figure 1 retrieval tasks typically use an embedding model to vectorize both the passage and the query, compute the similarity between the resulting embeddings, and then return the passage with the highest similarity as the matching result (Li et al., 2021). Although this approach is simple and easy to implement, the training dataset for its core component – the embedding model – is usually biased, i.e., consisting mostly of simple queries. This limits the ability of embedding model to handle complex, hard queries, leading to a serious performance bottleneck and an upper limit on its effectiveness.

To address hard queries, existing approaches can be broadly categorized into two types: query rewriting methods and model-centric methods. Query rewriting methods propose rewriting each query into simple one during the inference phase for ease of processing by the embedding model, such as BEQUE (Peng et al., 2024) and MiniELM (Nguyen et al., 2025). Though effective, such approaches may cause the loss of query semantics and fail to fully capture the retrieval intent of users. To address this, model-tuning methods improve the embedding models by designing novel training strategies. For example, BGE-M3-Embedding (Chen et al., 2024), Conan-Embedding (Li et al., 2024) uses a batch contrastive training strategy or its variants to optimize the embedding model, batch contrastive training strategy usually use in-batch samples as negatives to dynamically expand the original dataset, which may introduce false negatives. Although these methods have achieved

significant success, they still rely on inherently biased datasets, which limits the full potential of the embedding model from being fully realized.

Orthogonal to model-centric or query rewriting methods, this paper proposes addressing the hard query challenge from the data level by generating hard queries to improve the embedding model. More specifically, Considering that the generated queries need to possess semantic or logical knowledge similar to that of humans, and inspired by the ability of agents to closely simulate human behavior, this paper proposes a multi-agent framework to generate hard queries, thereby improving the training performance of the embedding model. The core idea of our approach is to first adopt a generation agent to create new queries. Then, two specialized agents, i.e., Code agent for logical reasoning and chain-of-thought(CoT) (Wei et al., 2022) agent for semantic understanding, evaluate the queries to identify truly hard ones. Finally, agents with expertise in different domains engage in a discussion to further optimize the selected hard queries. Experimental results demonstrate that our method outperforms existing approaches. The major contributions of this paper are summarized as follows:

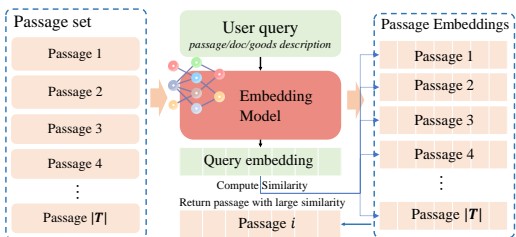

Figure 1: Retrieval framework.

- As is known to us, we are the first to propose a multi-agent data generation framework specifically for retrieval tasks. This framework leverages different agents to respectively handle the processes of generation, validation, and optimization, ensuring that the generated hard queries are close to real-world user queries.

- For each generated sample, we introduce a dual-agent comparative verification scheme. One agent performs logical and rule-based validation by executing code, while the other conducts semantic validation through chain-of-thought reasoning, enabling the effective identification of truly hard queries.

- We propose a multi-agent group discussion mechanism, involving agents with expertise across broader domains, to perform final validation and optimization of the hard queries, thereby ensuring the reliability of the generated samples.

- We conduct extensive experiments on different datasets and models. Experimental results show that our proposed model consistently outperforms existing methods.

## 2 RELATED WORK

**Hard Query Handling in Retrieval.** Processing hard queries are challenging due to their semantic complexity, low frequency, and lack of labeled data. Existing methods typically focus on model-centric improvements or query rewriting. On the model side, BGE M3-Embedding (Chen et al., 2024), Conan-Embedding (Li et al., 2024), SimCSE (Gao et al., 2021), RocketQA (Qu et al., 2020), RocketQAV2 (Ren et al., 2021), DPR(Karpukhin et al., 2020), NV-Embed Lee et al. (2025) improves embedding quality by dynamically mining hard negatives. Although these methods can improve the robustness of the representation, they focus primarily on better utilization of existing data rather than generating new supervised signals, what's more, in-batch negative samples inherently may be misleading for retrieval tasks, the dynamic sampled in-batch negatives may be the false negatives, which will result in performance drop. Another method of handling hard queries is query rewriting, query rewriting methods propose rewriting each input query into simple one during the inference phase such that the embedding model can process hard queries. For example, BEQUE (Peng et al., 2024) rewrites long-tail hard queries into head-form equivalents using instruction-tuned LLMs, while MiniELM (Nguyen et al., 2025) leverages knowledge distillation and reinforcement learning to adapt rewriting strategies from real-time feedback. Though effective, such approaches may cause the loss of query semantics and fail to fully capture the user's search intent. Orthogonal to model tuning or query rewriting methods, this paper proposes addressing the hard query challenge from the data level by generating hard queries to well train the embedding model, and the hard queries will push the upper boundary of an embedding model that can improve the generality.

**Multi-Agent Systems.** Recent research has explored the potential of multi-agent systems to enhance the reasoning capabilities of large language models across various domains. Multiagent Debate (Du et al., 2023) leverages agent debate mechanisms to enhance the factual accuracy of model outputs. GroupDebate (Liu et al., 2024) proposes a group-based agent discussion framework that enhances reasoning efficiency by facilitating intra-group deliberation and inter-group consensus. MathChat (Liang et al., 2024) presents a conversational multi-agent framework designed to tackle complex mathematical problems. Unlike these methods, our approach employs multi-agent discussion to refine labels for long-tail hard samples, enabling high-quality data generation for hard queries in retrieval tasks.

**Retrieval Model Training and Preference Alignment.** Traditional retrieval models, such as DPR (Karpukhin et al., 2020) and E5 (Wang et al., 2022), rely on manually labeled data or unsupervised contrastive learning (e.g., InfoNCE (van den Oord et al., 2018)) to learn query-passage interactions. Recent advancements integrate LLMs to improve retrieval quality by leveraging their contextual understanding. For example, Syntriever (Kim & Baek, 2025) proposes partial Plackett-Luce ranking, which combines preference modeling with contrastive learning to align retrievers with LLM-generated relevance judgments. Despite their advancements, the above methods still struggle with long-tail hard queries, which are infrequent but potentially highly relevant for certain users.

## 3 BACKGROUND

**Hard Samples.** In this work, we define hard samples as query-passage pairs that are non-standard, difficult to obtain, ambiguous, or require external knowledge for proper understanding and matching. These samples typically exhibit characteristics such as complex semantic structures, low frequency in training data, or the need for additional contextual knowledge beyond the immediate text. The distribution of generated hard queries is controlled through carefully designed prompts to LLMs, which instruct the models to simulate user behavior and generate more challenging queries that closely resemble real-world usage patterns.

**Dense Passage Retrieval.** This paper focus on optimizing the embedding model to improve the retrieval task performance. As shown in Figure 1, the retrieval task aims to retrieve the most relevant passages from the passage corpus $\mathcal{P}$ for any query $q$.

To achieve this, a pretrained embedding model $E(.;\mathbf{w})$ with the weights $\mathbf{w}$ is adopted to vectorize each passage $p_j \in \mathcal{P}$ and obtain $E(p;\mathbf{w})$. Then, it embeds the query $q_i$ to obtain $E(q_i;\mathbf{w})$ and calculates the distance between the query $q_i$ and passage $p_j$, i.e.,

$$d(q_i, p_j) = 1 - \cos(E(q_i;\mathbf{w}), E(p_j;\mathbf{w})). \tag{1}$$

where $\cos(\cdot, \cdot)$ denotes the cosine similarity between two embeddings. The retriever will finally return a passage $p_i^*$ that achieves the minimum distance among all passages $p \in \mathcal{P}$ corresponding to the query $q_i$, i.e., $p_i^* = \arg\min_{p \in \mathcal{P}} d(q_i, p)$. The retrieval task usually adopts a ground-truth label $y \in \{0, 1\}$ to denote whether the returned $p^*$ matches the query $q_i$, where $y = 1$ denotes a match between the returned result and the ground truth passage, whereas denotes a mismatch. During the training phase, the retrieval task seeks to minimize the following contrastive loss of all queries in the training dataset $q \in \mathcal{T}$ to train the embedding model $E(.;\mathbf{w})$:

$$\min_{\mathbf{w}} \mathcal{L}(\mathbf{w}) = \frac{1}{|\mathcal{T}|} \sum_{q_i \in \mathcal{T}} (y_i \cdot d(q_i, p_i^*)^2 + (1 - y_i) \cdot \max(0, m - d(q_i, p_i^*))^2), \tag{2}$$

where $m$ is a hyperparameter that sets the margin (typically $m = 0.5$).

**Objective of this paper.** Although the above equation (2) can be used to obtain an embedding model, the model usually suffers from poor performance because the hard query samples in the training dataset follow a long-tailed distribution, and the model cannot be sufficiently trained on these hard samples. To address this issue, we propose in this paper to increase the number of hard samples through sample generation. While some traditional rule-based methods for data augmentation or generation already exist, they often fail to generate queries that are close to real human queries,

potentially losing the rich underlying semantics presented in genuine user inputs. Recently, LLM-based agents have attracted significant attention because they can closely mimic human speaking styles, language content, and expression patterns, etc. Inspired by this, we propose to generate samples using an agent-based approach, aiming to alleviate the problem of the long-tailed distribution of hard samples. Formally, we aim to generate a new query set $\mathcal{T}'$ from the query training set $\mathcal{T}$ based on LLM agents.

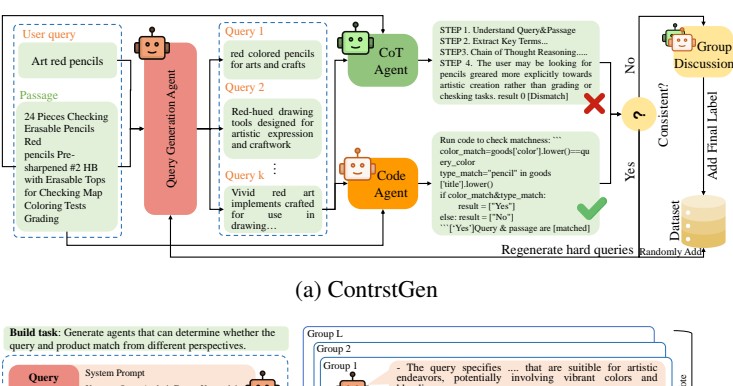

(a) ContrstGen

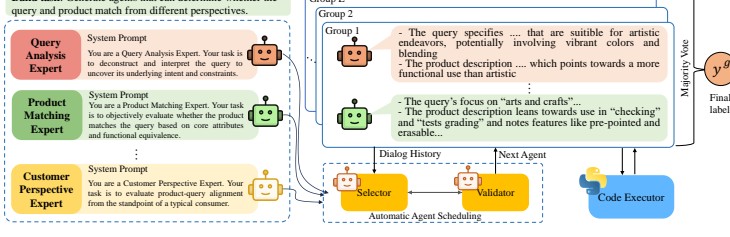

(b) Group Discussion

Figure 2: Overall framework of ContrastGen Framework is shown in (a): Given the original query and a candidate passage, QueryGen iteratively generates hard queries until the matchness between the generated query and the passage are inconsistent among Code and CoT agents, indicating a hard sample. For the hard sample, we use multi-agent group discussion to determine the final label. Group Discussion Framework is depicated in (b): Given a task description, agents are built based on task-specific prompt and then they conduct multi-turn discussions from different perspectives scheduled by a nested two-agent chat. If code verification is needed, a Code Executor is invoked to assist the reasoning process. Multiple groups independently analyze the query–passage pair and vote to determine the final consensus. Group Discussion is a part of ContrastGen Framework mainly used for label refinement.

## 4 METHOD

This section introduces ContrastGen, a multi-agent based retrieval data generation framework. As shown in Figure 2a, ContrastGen consists of two stages, i.e., contrastive agents based hard sample generation and group discussion based label refinement. The sample generation stage iteratively generates hard queries and the multi-agent group discussion stage determines the final label.

### 4.1 STAGE 1. CONTRASTIVE AGENTS BASED HARD SAMPLE GENERATION

This stage aims to iteratively generate hard samples. Specifically, we first adopt a generation agent named QueryGen to rewrite the query. Further, considering that hard samples are often easily interpreted by agents with different perspectives as matching different passages, we further leverage two distinct agents – the Code Agent and the CoT Agent – to contrastively assess the matchness between the generated query and the original passage. When these two agents produce conflicting matching results, the corresponding sample is identified as a hard sample and is then fed into the subsequent discussion stage to determine its final matching label. Otherwise, the QueryGen agent is invoked to regenerate samples until a hard sample is generated.

**Iterative query generation via QueryGen agent.** The QueryGen Agent iteratively synthesizes challenging queries based on the given passage and historical queries. In each iteration $k$, ContrastGen feeds the query-passage pair $(q_i, p_i)$, along with the set of previously generated queries by the QueryGen agent $\mathcal{E}_i = \{q'_{i,1}, q'_{i,2}, \ldots, q'_{i,k-1}\}$, and a prompt (see Appendix A.1) into the QueryGen agent $Q(\cdot)$, to generate a new query $q'_{i,k} = Q(q_i, p_i, \mathcal{E}_i)$. Then, ContrastGen forms a pair $(q'_{i,k}, p_i)$ by combining $q'_{i,k}$ with the passage $p_i$, and feeds this pair into both the Code Agent and the Chain-of-Thought (CoT) Agent. If both agents consistently judge that $(q'_{i,k}, p_i)$ match or mismatch each other, the query is considered *easy*, and it is added to the historical query list $\mathcal{E}_i = \mathcal{E}_i \cup \{q'_{i,k}\}$. With a sampling probability $r$, the query is also included in the generated training dataset $\mathcal{T}' = \mathcal{T}' \cup \{q'_{i,k}\}$, or equally, the query is discarded with a probability $1 - r$. The above process is then repeated to generate a new query. Notably, $\mathcal{T}'$ is initialized as $\mathcal{T}$. However, if the Code Agent and the CoT Agent disagree on whether $(q'_{i,k}, p_i)$ match, the query is considered *hard*, and the generation process stops.

**Contrastive matchness assessment via Code and CoT agents.** After obtaining the generated query, ContrastGen employs both the Code Agent and the CoT Agent to verify whether the query matches the original passage. The Code Agent assesses the match based on structured rule-based reasoning, while the CoT Agent determines the match through semantic understanding. Therefore, these represent two fundamentally different perspectives. When the query is complex and difficult to understand, it is more likely to lead to inconsistent judgments from the two agents.

*Code agent.* By taking the query-passage pair $(q'_{i,k}, p_i)$ as input, the Code Agent will extract several attributes or expressions that can be compared or computed, and then automatically generate code to perform reasoning through program execution in a sandbox environment. The prompt and execution example are shown in Appendix A.2. Finally, the Code Agent $C_d(\cdot)$, by invoking the execution engine, outputs a matching decision result $y^d_{i,k} = C_d(q'_{i,k}, p_i)$ indicating whether $q'_{i,k}$ and $p_i$ match.

*CoT agent.* To assess the matching of the query-passage pair $(q'_{i,k}, p_i)$, CoT agent applies the chain-of-thought reasoning with four steps (an example and prompts can be seen in Appendix A.3): 1). Understand the query and the passage. Clarify user intent and identify product features. 2). Extract Key Terms. Highlight key attributes, synonyms, and functional components. 3). Apply Reasoning. Compare extracted query terms with passage attributes, accounting for synonyms and logical equivalences. 4). Generate a Conclusion. Decide on match/mismatch based on accumulated evidence.

With the four reasoning steps, the CoT agent $C_t(\cdot)$ obtains the match results $y^t_{i,k} = C_t(q'_{i,k}, p_i)$. Finally, ContrastGen checks whether $y^d_{i,k}$ and $y^t_{i,k}$ are the same. If they match, the sample is considered an easy sample, and it is fed back to QueryGen to generate a new query. The sample is also added to the generated dataset with a certain probability. Otherwise, the process proceeds to the next stage, where a group discussion mechanism is employed to make the final determination on whether the pair matches.

## 4.2 STAGE 2. MULTI-AGENT GROUP DISCUSSION FOR LABEL REFINEMENT

To determine the final matching label, the hard query is delivered to the multi-agent group discussion stage, where multiple groups independently analyze the query-passage pair and vote to reach the final consensus.

*Multi-agent group building.* For each group, we build discussion agents based on task-specific expertise roles, simulating a realistic multi-role discussion scenario, as illustrated in Figure 2b. We use various system prompts to build different expert agents. For instance, when the passage are products, the agents are built by including the Query Analysis Expert, Product Matching Expert, Customer Perspective Expert, and Market Trends Expert, etc. An example can be found in Appendix A.4. These agents engage in role-based multi-turn discussions, each focusing on their specialized perspective to validate or refute the initial matching. Besides, we allow agents to invoke executable code snippets or perform commands via a code executor, when logic verification is required.

*Discussion.* To coordinate the discussion among multiple agents within each group $l$, we adopt an automatic agent scheduling strategy which includes an Agent Selector and a Validator described in (Wu et al., 2023). The Selector proposes the name of the next agent to speak based on the dialogue history, while the Validator checks whether the proposal corresponds to a valid agent. If the proposed

name does not match any existing agent, the Validator feeds this back to the Selector, prompting it to generate a new proposal. This process repeats until a valid agent speaker is identified or the maximum number of attempts is reached. If all attempts fail, a fallback strategy–such as round-robin–is used. In each round $t$, the selected agent $A_t(\cdot)$ takes as input the query-passage pair $(q'_{i,k}, p_i)$ and the historical decisions from other agents in the group:

$$\mathcal{A}_t = \left\{ A_1(q'_{i,k}, p_i), A_2(q'_{i,k}, p_i, A_1), \ldots, A_{t-1}(q'_{i,k}, p_i, \mathcal{A}_{t-1}) \right\} \tag{3}$$

and produces a decision $A_t(q'_{i,k}, p_i, \mathcal{A}_t)$. After $T$-round discussion among agents, ContrastGen applies a majority voting scheme over all agents' decisions to produce the final group decision:

$$y_1^g = \text{MajorityVote}(A_1, A_2, \ldots, A_T). \tag{4}$$

Next, our empirical study shows that the very first decision that the CoT agent made elicits more improvement than directly use majority voting over only $L$ independent groups, while significantly reducing token consumption at the same time. Therefore. ContrastGen makes the final decision by applying a global greedy voting process from $L$ independent groups' decision and previous CoT agent's decision denoted by $y_0^g$, This approach is adopted because majority voting may introduce majority bias and cannot handle situations with an even number of groups: if half of the groups agree and the other half disagree, it becomes impossible to reach a final decision. The decision process can be defined as:

$$y^g = \left\{ \begin{array}{ll} y_1^g & y_0^g \neq y_1^g = y_2^g = \ldots = y_L^g \\ y_0^g & otherwise \end{array} \right.$$

Based on the value of $y^g$, ContrastGen determines the final matching label for the query-passage pair. Specifically, $y^g = 1$ signifies a matching pair, which is labeled as a positive example, while $y^g = 0$ denotes a non-matching pair, treated as a negative example. Finally, the embedding model is trained using the generated dataset $\mathcal{T}'$ with equation (2).

## 5 EXPERIMENTS

### 5.1 EXPERIMENT SETUP

**Datasets.** We evaluate our approach on three publicly available datasets: the *Shopping Queries* (Reddy et al., 2022), the *arXiv* (Clement et al., 2019), and *MS MARCO*(Nguyen et al., 2016). The Shopping Queries Dataset consists of multilingual search queries, each associated with up to 40 related products, annotated using ESCI relevance judgments (Exact, Substitute, Complement, Irrelevant). The arXiv Dataset contains over 1.7 million scholarly articles, including metadata such as titles and authors. For this dataset, queries were generated using GPT-4o-Mini (Hurst et al., 2024). MS MARCO is a large-scale dataset designed for training and evaluating models on tasks like reading comprehension, passage ranking, and question answering. ALL datasets utilize binary relevance labels (0 or 1) to indicate query-passage matches, facilitating an assessment of search quality and user experience enhancement. We implement ContrastGen based on the LangChain[1] and AutoGen (Wu et al., 2023) frameworks, and use these two datasets to generate new contrastive data.

**Baseline.** We compare our approach of training models with incrementally generated data against several baseline methods. *BM25* (Robertson et al., 2009): A traditional keyword-based retrieval model employing the BM25 ranking function. *Zero-Shot*: A pre-trained model evaluated without task-specific training or fine-tuning on the target dataset. *Original Data*: A model trained solely on the original dataset without data augmentation. *In-Batch Negatives* : A widely used method in SimCSE (Gao et al., 2021), RocketQA (Qu et al., 2020), RocketQAV2 (Ren et al., 2021), DPR(Karpukhin et al., 2020) that dynamically selects negative samples from the same training batch to enhance discriminative capability. CoT (Wei et al., 2022): a commonly used method in problem solving of large language models that use a series of intermediate reasoning steps to improve the ability of

---

[1]`https://github.com/langchain-ai/langchain`

Table 1: Performance comparison of different models and training methods on multiple datasets. The best results for each PELM on each dataset are highlighted in bold.

| Method | PELM | Shopping | | | arXiv | | | PELM | MS MARCO | | |
|---|---|---|---|---|---|---|---|---|---|---|---|
| | | R@10 | P@10 | N@10 | R@10 | P@10 | N@10 | | R@10 | P@10 | N@10 |
| BM25 | | 0.4664 | 0.3145 | 0.5115 | 0.369 | 0.037 | 0.2768 | | 0.6921 | 0.0784 | 0.4827 |
| Zero Shot | | 0.4617 | 0.3156 | 0.5165 | 0.3527 | 0.0342 | 0.2619 | | 0.8033 | 0.0908 | 0.5399 |
| Original Data | | 0.464 | 0.3182 | 0.5187 | 0.3615 | 0.0358 | 0.2743 | | 0.7771 | 0.0876 | 0.527 |
| COT | Conan | 0.4481 | 0.313 | 0.5148 | 0.3482 | 0.0349 | 0.2153 | Conan | 0.8036 | 0.0908 | 0.5386 |
| In-Batch Neg | | 0.4816 | 0.3364 | 0.5458 | 0.3732 | 0.0371 | 0.2895 | | 0.7998 | 0.0902 | 0.5397 |
| ContrastGen | | **0.4961** | **0.3494** | **0.5534** | **0.3859** | **0.0386** | **0.3021** | | **0.8132** | **0.0918** | **0.5427** |
| Zero Shot | | 0.5329 | 0.3566 | 0.5896 | 0.5886 | 0.059 | 0.355 | | **0.9321** | 0.105 | 0.6409 |
| Original Data | | 0.5362 | 0.3857 | 0.6073 | 0.6087 | 0.061 | 0.4346 | | 0.9293 | 0.105 | 0.6426 |
| COT | BGE-M3 | 0.5381 | 0.3792 | 0.6021 | 0.5995 | 0.06 | 0.3779 | BCE | 0.929 | 0.1048 | 0.641 |
| In-Batch Neg | | 0.542 | 0.3909 | 0.6048 | 0.5978 | 0.0599 | 0.3685 | | 0.9192 | 0.1037 | 0.6343 |
| ContrastGen | | **0.5565** | **0.4104** | **0.6362** | **0.6179** | **0.0618** | **0.445** | | **0.9321** | **0.1051** | **0.6428** |

large language models to perform retrieval data labeling. In this experiment, we use gpt-4o-mini as the base large language model for CoT prompting, generating 3 queries for each candidate passage. These baselines serve as reference points to evaluate the effectiveness of our proposed incremental data generation strategy.

**PELM.** Our study employs three pretrained embedding language model (PELM): *BGE-M3* model (Chen et al., 2024), *Conan-embedding-v1* (Conan) model (Li et al., 2024) and *bce-embedding-base_v1* (BCE) model (Youdao, 2023). BGE-M3 is a retrieval model trained on diverse datasets and excels in multilingual and cross-lingual tasks, accommodating inputs ranging from short queries to documents of up to 8,192 characters. Conan-embedding-v1 further enhances embedding quality through dynamic hard negative mining and cross-GPU balancing loss, which expose the model to more informative negative samples during training and alleviate memory constraints. Since BGE-M3 is pretrained in MS MARCO dataset, so we use BCE as a replacement to evaluate the effectiveness of our method, bce-embedding-base_v1 is a bilingual and crosslingual embedding model, excelling in Chinese, English and their crosslingual retrieval task.

**Evaluation Metrics.** We assess model performance using Precision@K (P@K), Recall@K (R@K), and NDCG@K (N@K), with $K = 10$. Precision@10 quantifies the proportion of relevant items among the top 10 retrieved results, Recall@10 measures the proportion of relevant documents retrieved within the top 10, and NDCG@10 evaluates ranking quality by assigning higher scores to correctly ranked relevant items. The test set consists of 10,000 items sampled randomly from both original and generated data.

**Configurations.** Unless otherwise specified, models are trained with a learning rate of $2e - 6$ for the Shopping Queries dataset, $1e - 6$ for the arXiv and MS MARCO dataset. Training data comprises 2,000 original samples and 2,000 additional generated samples, totaling 4,000 samples. In-batch negative sampling is conducted using 2,000 samples by default. For MS MARCO dataset, we use BM25 method and BCE embedding model to filter out the datasets that the ground truth answer are present in the top 10 candidate passages, queries that are longer than 8 are also filtered out, and we finally sample 5000 examples as the the training dataset, 2000 examples as the test set. All experiments were conducted on a single RTX 4090 GPU with 24 GB of VRAM, a 16-core Intel Xeon(r) Gold 6430 CPU, and 120 GB of RAM.

## 5.2 PERFORMANCE OVERVIEW

**Performance Analysis.** On average, each hard sample requires approximately 58 API calls, consumes 102,800 tokens (70,000 input and 32,800 output tokens), and completes within 30 seconds, with computational time distributed across CoT reasoning (2-3s), code generation (8-12s), and group discussion phases (15-20s).

**Retrieval Performance.** Table 1 presents the overall retrieval performance of different methods across the Shopping Queries, arXiv, and MS MARCO datasets. ContrastGen achieves the best overall results, reaching a Recall@10 of 55.65% on Shopping Queries, 61.79% on arXiv, and 93.21% on MS

Table 2: Performance with different data types (Easy, Hard, Mixed).

| Data Type | Shopping | | | arXiv | | |
|---|---|---|---|---|---|---|
| | R@10 | P@10 | N@10 | R@10 | P@10 | N@10 |
| Easy | 0.5110 | 0.3718 | 0.5854 | 0.6087 | 0.0610 | 0.4289 |
| Hard | **0.5361** | **0.3766** | **0.5981** | **0.6162** | **0.0617** | **0.4385** |
| Mix | 0.5327 | 0.3742 | 0.5973 | 0.6095 | 0.0610 | 0.4352 |

MARCO, outperforming both the vanilla BGE-M3, Conan models and BCE embedding models. The conan model shows consistent improvements when incorporating generated data, but they still lag behind BGE-M3 and BCE across all metrics. Notably, all models benefit from generated training data, indicating the generalizability of ContrastGen.

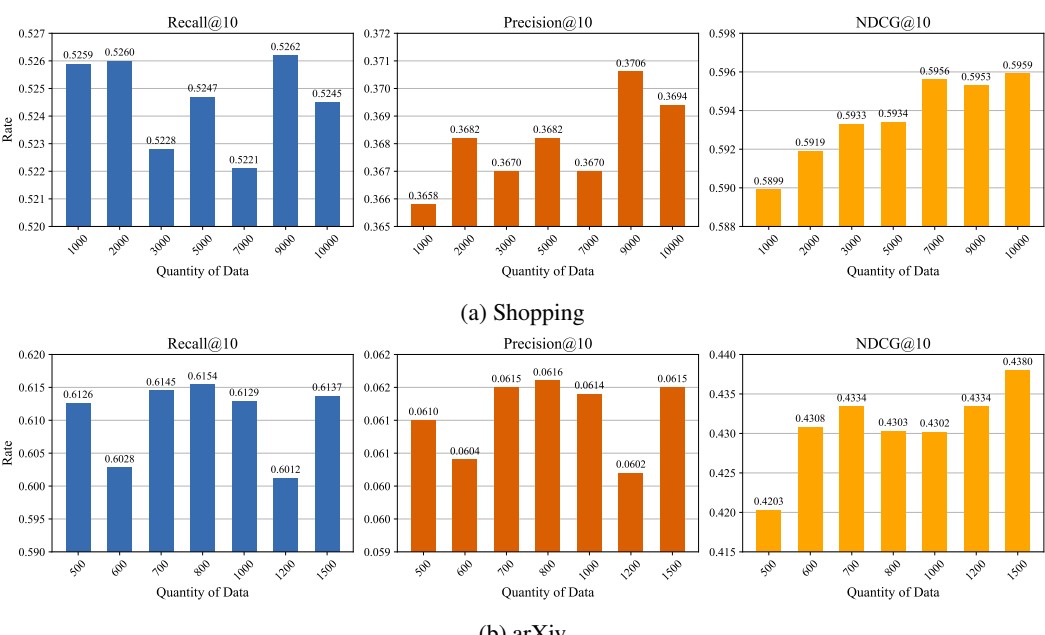

(a) Shopping

(b) arXiv

Figure 3: Performance evaluation using different generated data quantities. The abscissa denotes the quantity of data, while the ordinate represents the Metric Value.

**Effect of Data Quantity.** We use Shpping Queries and arXiv dataset to investigate the effect of data quantity. As shown in Figure 3, retrieval performance improves with moderate data augmentation but degrades when exceeding an optimal threshold. For the Shopping Queries dataset 3a, Recall@10 and Precision@10 peak at 9,000 samples (0.5262 and 0.3706), while NDCG@10 continues to rise slightly to 0.5959 at 10,000 samples. In contrast, the arXiv dataset 3b shows peaks in Recall@10 (0.6154) and Precision@10 (0.0616) at 800 samples, but NDCG@10 improves further to 0.4380 at 1,500 samples. This non-monotonic trend suggests that excessive data introduces noise or redundancy, reducing model generalization. The optimal data quantity varies by task: $9,000-10,000$ samples work best for Shopping Queries, while arXiv benefits most from 1,500 samples. Thus, data augmentation should be tailored to the dataset to balance performance gains and overfitting risks.

**Influence of Data Type.** Table 2 investigates the effect of different data types (Easy, Hard, Mixed) on model performance across two tasks: Shopping Queries and arXiv. Hard samples consistently yield the highest performance across all metrics (Recall@10, Precision@10, NDCG@10) in both tasks. For Shopping Queries, Hard data achieves 0.5361 (Recall@10), 0.3766 (Precision@10), and 0.5981 (NDCG@10); for arXiv, these values are 0.6162, 0.0617, and 0.4385 respectively. This indicates that challenging samples effectively enhance the model's discriminative ability to distinguish between positive and negative instances.

**Ablation study.** To evaluate the contribution of each individual component of ContrastGen, we compare three distinct configurations: (1) CoT Agent only, where only the CoT agent is employed

Table 3: Ablation study. The first three columns represent the following components: CoT agent, Code agent, Multi-agent group discussion. R@10, P@10 and N@10 respectively represent Recall@10, Precision@10 and NDCG@10.

| CoT | Code | Discussion | Shopping | | | arXiv | | |
|-----|------|-----------|------|------|------|------|------|------|
| | | | R@10 | P@10 | N@10 | R@10 | P@10 | N@10 |
| ✓ | ✗ | ✗ | 0.5268 | 0.3659 | 0.5825 | 0.6154 | 0.0616 | 0.4292 |
| ✗ | ✓ | ✗ | 0.5268 | 0.3662 | 0.5915 | 0.6137 | 0.0615 | 0.4375 |
| ✓ | ✓ | ✓ | **0.5374** | **0.3770** | **0.6060** | **0.6179** | **0.0619** | **0.4450** |

for labeling queries based on semantic reasoning; (2) Code Agent only, where only the Code agent is used to label queries based on logical and rule-based validation through executable code; and (3) ContrastGen, which integrates all components. The performance of each configuration in terms of data quality and downstream retrieval performance is summarized in Table 3. The results clearly indicate that removing any component leads to a measurable decline in performance. When only the CoT agent is used, the generated queries may suffer from hallucinations or semantic drifts due to the generative nature of the model. On the other hand, relying solely on the Code agent may miss nuanced linguistic variations and fail to capture complex user intents, leading to overly rigid or syntactically correct but semantically less meaningful queries. In contrast, the full ContrastGen framework benefits from the complementary strengths of both agents. Discrepancies between the CoT and Code agent outputs are effectively resolved through the multi-agent group discussion mechanism, which brings in diverse perspectives and domain expertise to refine and validate the final label of hard samples. This collaborative filtering and refinement process significantly improves label accuracy and ensures higher fidelity in the resulting query-passage pairs. Overall, this ablation study highlights the necessity of each component and demonstrates that their combination is crucial for achieving optimal performance.

**Case study.** Table 4 presents two examples of the ContrastGen-enhanced data, including the original query, the generated easy and hard queries, and their corresponding matching labels with the passage.The first example illustrates the process for generating a hard positive sample: the original positive query "black baseball cap" is gradually elaborated through ContrastGen's iterative generation, resulting in the hard positive query "unstructured relaxed fit black cap for all seasons". While the second example shows the process for generating a hard negative sample: the original negative query "6 quart crockpot" is similarly refined through iterative generation, producing the hard negative query "soft queen blankets for winter warmth". This table provides an intuitive illustration of the diversity and complexity of the generated data, further validating the effectiveness of ContrastGen in producing high-quality hard samples.

Table 4: The instance of ContrastGen enhanced data

| Passage | Type | Query | Label |
|---------|------|-------|-------|
| Distressed Baseball Cap - Mom Life (Black) Vintage style; Washed & distressed; Low profile crown. Unconstructed style gives off a "dad hat" vibe. Suitable for all seasons. Features: adjustable closure, unstructured crown, all-day comfort. | Original | black baseball cap | 1 |
| | Easy | vintage unstructured dad hat for moms | 1 |
| | Hard | unstructured relaxed fit black cap for all seasons | 1 |
| Mellanni Queen Sheet Set - Hotel Luxury 1800 Bedding. Extra soft cooling sheets & pillowcases; deep pocket (up to 16 inch mattress). Wrinkle, fade, stain resistant (4 Piece, Queen, White). | Original | 6 quart crockpot | 0 |
| | Easy | cool beds for boys | 0 |
| | Hard | soft queen blankets for winter warmth | 0 |

Fig 4 shows an example of ContrastGen, The full reasoning process is too long, so we omit the less important parts of reasoning, leaving the key reasoning process, CoT Agent in Fig 4a obtains the final answer by using the predefined step by step reasoning, while the Code Agent in Fig 4b obtains the final answer by using the executed the Python Code. the result of the Cot Agent is 1, but the Code

Agent's output is "No" (which means 0, or "not relevant"). So the query('Obi-Wan Kenobi strategy game expansion') and the passage can be relatively hard to distinguish (labeled as hard).

# 6 CONCLUSION AND FUTURE WORK

To address the challenge of insufficient hard queries in training datasets for embedding models in retrieval tasks, this paper proposes ContrastGen, which leverages a multi-agent framework to generate hard samples. The core idea of the method is to employ different agents to separately handle the generation, discrimination, and optimization of hard queries, ultimately forming effective positive or negative hard query-passage pairs. Experiments conducted on various types of embedding models and different datasets demonstrate that our approach consistently outperforms existing methods.

Although the proposed ContrastGen framework achieves promising results in generating high-quality hard queries for retrieval tasks, the proposed method remains unrelated to the embedding model. Therefore, we aim to incorporate feedback mechanisms from the embedding model itself into the query generation pipeline. By enabling a closed-loop training paradigm, where the model performance on hard queries guides further data generation, we can achieve more targeted and iterative improvements in model robustness.

**CoT Agent** ✓

**Step 1: Understand the Query and Passage**
The query is focused on an Obi-Wan Kenobi strategy game expansion. The key Passage features include that it is an expansion for the Star Wars Imperial Assault game, requires the core set to play, involves tactical combat in the Star Wars universe, particularly highlighting Obi-Wan Kenobi as a character, and offering new missions and customization options.
**Step 2: Extract Key Terms**
- Query Terms: Obi-Wan Kenobi, strategy game, expansion
- Passage Features:
- Expansion for Star Wars Imperial Assault
- Requires Imperial Assault Core Set
- Tactical combat in Star Wars universe
- Focus on Obi-Wan Kenobi
- New missions and customization
**Step 3: Apply Chain-of-Thought Reasoning**
1. The Passage is clearly labeled as an expansion for the Star Wars Imperial Assault game, which aligns with the query's focus on a strategy game expansion.
2. The description specifically mentions "Obi-Wan Kenobi" as a key character within this expansion, confirming the relevance.
3. The nature of gameplay (tactical combat) aligns with the strategic aspect implied in the query, indicating that user intent is met.
4. Additional features such as new missions and customization options enhance the gameplay, which is typical in game expansions and supports user interest in strategic elements.
**Step 4: Generate Conclusion**
The passage matches the query regarding the focus on Obi-Wan Kenobi and being a strategy game expansion related to Star Wars Imperial Assault.
FINAL RESULT: [1]

**Code Agent** ✗

The key idea to solve the problem is to compare the user query with the provided passage description and determine whether the product fits the criteria outlined in the rules. ..... Now, I will write the Python code to reflect this determination.

```python
# Determine if the passage description
    matches the user query
query = "Obi-Wan Kenobi strategy game
    expansion"
passage_description = {
    "title": "Star Wars Imperial Assault
        Board Game .....",
    "bullet_point": ".... With a new
        campaign side mission, two new
        skirmish missions, and new Deployment
        cards, Item cards.....",
    "brand": "Fantasy Flight Games",
    "color": None
}
is_expansion_required = "a copy of the
    Imperial Assault Core Set is required
    to play" in passage_description["
    bullet_point"]
if ("Obi-Wan Kenobi" in passage_description[
    "title"] and
    "strategy game" in passage_description["
    title"] and
    "expansion" in passage_description["
    bullet_point"]):
    if is_expansion_required:
        print(["No"])
    else:
        print(["Yes"])
else:
    print(["No"])
```

Code executed result: ['No']

(a) CoT Agent                    (b) Code Agent

Figure 4: Case study of ContrastGen. The output differences between the CoT Agent and Code Agent indicate that the query 'Obi-Wan Kenobi strategy game expansion' is complex for this passage and is difficult to distinguish, so we label this query and passage as a hard data that will be valuable for improving the performance of text embedding retrieval.

## 7 REPRODUCIBILITY STATEMENT

To support the reproducibility of our work, we provide the following information:

- Code: We will release our source code under an open-source license (e.g., MIT License), and make it public available if this paper is accepted.
- Data: Our experiments use publicly available datasets: the Shopping Queries dataset, the arXiv and MS MARCO dataset, all the processing details are described in this paper.
- Hyperparameters: The core hyperparameters are described in section 5, we provide the primary training script in our source code.
- Software Environment: We provide a requirements.txt file to reproduce our software environment.
- Computational Resources: Our primary experiments were run on a single RTX 4090 GPU with 24 GB of VRAM described in section 5.

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

# APPENDIX

## A  KEY PROMPTS

### A.1  PROMPT FOR QUERYGEN AGENT

**Positive QueryGen (GPT-4o-mini)**

```
1  # You are an expert in information retrieval, and I need you to generate
       several new queries based on the product information and the
       historical queries provided.
2
3  ## Requirements for generated queries:
4  - Each query should be no more than eight words.
5  - Each query should be more challenging to match with the product than
       previous queries.
6  - The language of the output content must be English.
7
8  ## Provided inputs:
9  - Product information: {{product}}
10 - Historical queries: {{queries}}
11 - Number of queries to generate: {{num}}
12
13 ## Output format (JSON):
14 {
15    "queries": ["generated_query_1", "generated_query_2", "..."]
16 }
```

**Negative QueryGen (GPT-4o-mini)**

```
1  # You are an information retrieval expert, and I need help generating
       queries that do not match products.
2
3  ## Requirements:
4  - Each query should be no more than eight words.
5  - Each query should be more challenging to determine if it matches the
       product than previous queries.
6  - The language of the output content must be English.
7
8  ## Provided Inputs:
9  - Product information: {{product}}
10 - Historical queries: {{queries}}
11 - Number of queries to generate: {{num}}
12
13 ## Output format (JSON):
14 {
15    "queries": ["generated_query_1", "generated_query_2", "..."]
16 }
```

### A.2  PROMPT FOR CODE AGENT

**Summary (GPT-4o-mini)**

```
1  Summarize the takeaway from the conversation. Do not add any introductory
       phrases. Your response must be in JSON format:
2
3  {
4    "comments": "give a short explaination about the final decision"
5    "response_number": "<the final decision: 1(match) or 0(mismatch)>"
6  }
```

**Test Task (GPT-4o-mini)**

```
1  Let's use Python to solve a query goods' description matching problem.
2  Query requirements:
3  You should always use the 'print' function for the output.
4  You must follow the formats below to write your code:
5  ```python
6  # your code
7  ```
8  First state the key idea to solve the problem. You may choose from three
       ways to solve the problem:
9  Case 1: If the problem can be solved with Python code directly, please
       write a program to solve it.
10 Case 2: If the problem is mostly reasoning, you can solve it by yourself
       directly.
11 Case 3: If the problem cannot be handled in the above two ways, please
       follow this process:
12 1. Solve the problem step by step (do not over-divide the steps).
13 2. Take out any queries that can be asked through Python (for example,
       any calculations that can be calculated).
14 3. Wait for me to give the results.
15 4. Continue if you think the result is correct. If the result is invalid
        or unexpected, please correct your query or reasoning.
16 After you get the answer, put the answer in [] and reply TERMINATE.
```

**Problem (GPT-4o-mini)**

```
1  problem: given a user query and a product retrieved for this query, the
       goal of this task is to classify each product as match or mismatch
       for the query.
2  user query: {{query}}
3  goods description: {{description}}
4  rules:
5  - match: the goods is relevant for the query, and satisfies all the query
        specifications (e.g., water bottle matching all attributes of a
       query "plastic water bottle 24oz", such as material and size)
6  - mismatch:
7  1. the goods is somewhat relevant: it fails to fulfill some aspects of
       the query but the item can be used as a functional substitute (e.g.,
       fleece for a "sweater" query)
8  2. the goods does not fulfill the query, but could be used in combination
        with an exact item (e.g., track pants for a "running shoe" query)
9  3. the goods is irrelevant, or it fails to fulfill a central aspect of
       the query.
10
11 please return ["Yes"] if the goods description match the query, and ["No
       "] if they do not match.
```

### A.3    PROMPT FOR COT AGENT

**System (GPT-4o-mini)**

```
1  You are a CoT reasoner who can solve problems through a step-by-step
       thinking approach.Return 'TERMINATE' when the task is done.
```

**User (GPT-4o-mini)**

```
1  Query: {{query}}
2  Product: {{passage}}
3  Task: Determine if the query matches the product using reasoning steps.
4  Return 1 for match, 0 for mismatch.
5  Step 1: Understand the Query and Product
6  Identify the user's intent and key product features.
7  Step 2: Extract Key Terms
```

```
8  Extract relevant keywords, synonyms, and domain-specific terms.
9  Step 3: Apply Chain-of-Thought Reasoning
10 Analyze relationships between query terms and product attributes.
11 Step 4: Generate Conclusion
12 FINAL RESULT: [ ]
```

## A.4 PROMPT FOR DISCUSSION AGENT

**Discussion Task (GPT-4o)**

```
1  TASK: Determine if the query adequately describes the product's core
      features.
2  QUERY: {{query}}
3  PRODUCT: {{passage}}
4  INSTRUCTIONS:
5  1. Analyze the relationship between query and product
6  2. Discuss from different perspectives and allow semantic variations (e.g
      ., synonyms, different phrasing)
7  3. Consider logical equivalences (e.g., 'waterproof' vs 'can be used
      underwater')
8  4. Reach a consensus considering comprehensive coverage
9  5. Check scope inclusion: if the product's features fully cover the query
       requirements, it is match
10 6. Provide a binary output:
11 - Return 1 if query matches product
12 - Return 0 if query does not match product
13 The final generated content can only be 0 or 1 and do not generate
      anything else.
```

## B ADDITIONAL EXPERIMENTS

**Impact of Positive-Negative Ratios.** Table 5 examines the influence of different positive-to-negative sample ratios on model performance. Optimal performance is observed when the ratio is balanced within a specific range, whereas an excessive number of negative samples adversely affects model effectiveness. Notably, a 10:9 ratio yields the highest Recall@10 (0.5353) on Shopping Queries, whereas an imbalanced 1:2 ratio leads to reduced NDCG scores.

**Effectiveness of Generated Data under In-Batch Negatives Training.** Figure 5 compares BGE-M3 performance trained with varying quantities of original and generated data using the in-batch negatives strategy. Across all scales, models trained on generated data consistently outperform those using original data. For instance, with 500 samples, generated data achieves a Recall@10 of 0.5555 compared to 0.5365 for original data. This advantage persists even as data volume increases, indicating greater robustness and generalization. Additionally, performance with original data slightly declines as quantity increases, suggesting redundancy or domain saturation. These results highlight the effectiveness of high-quality synthetic data in retrieval tasks under contrastive training settings.

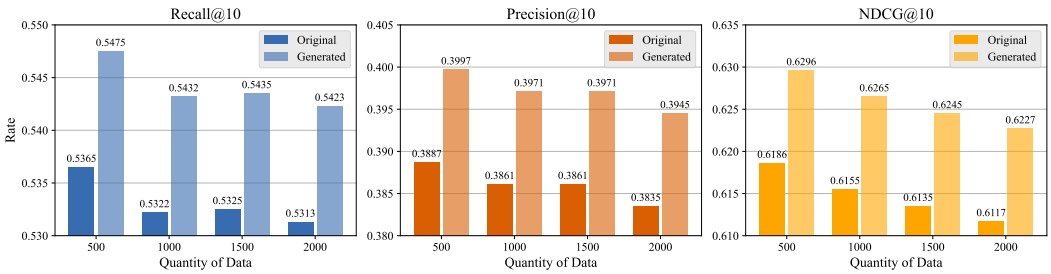

Figure 5: Performance comparison of the BGE-M3 model with original and generated data of different scales under In-Batch Negatives training.

Table 5: Impact of positive-negative sample ratios on performance.

| Ratio | Shopping | | | arXiv | | |
|---|---|---|---|---|---|---|
| | R@10 | P@10 | N@10 | R@10 | P@10 | N@10 |
| 0 | 0.5302 | 0.3894 | 0.5937 | 0.6104 | 0.0611 | 0.4321 |
| 10:2 | 0.5242 | 0.3882 | 0.5961 | 0.6237 | 0.0625 | 0.4193 |
| 10:3 | 0.5255 | 0.3870 | 0.5920 | 0.6171 | 0.0618 | 0.4153 |
| 10:4 | 0.5232 | 0.3870 | 0.5906 | 0.6171 | 0.0618 | 0.4192 |
| 10:5 | 0.5272 | 0.3882 | 0.5927 | 0.6179 | 0.0619 | 0.4205 |
| 10:9 | **0.5353** | **0.3966** | **0.5957** | **0.6254** | **0.0626** | 0.4271 |
| 1:1 | 0.5213 | 0.3894 | 0.5911 | 0.6212 | 0.0622 | **0.4335** |
| 1:2 | 0.5265 | 0.3869 | 0.5942 | 0.6112 | 0.0612 | 0.4193 |

## C  LIMITATIONS

**Limitation.** Our work mainly generates hard sample data through agent-based iterative processes. When a query is difficult to rewrite into a harder version, the number of iterations may reach the preset upper limit, leading to significant consumption of computational resources. Moreover, our approach employs a multi-agent system based on large language models to perform tasks such as generation, validation, and optimization, which also incurs substantial resource costs. Therefore, reducing the computational overhead of hard query generation is a key limitation that we aim to address in future work.

