# OpenReview forum: "ContrastGen: A Multi-agent Contrastive Framework for Hard Retrieval Data Generation"
_ICLR.cc/2026/Conference — Submitted to ICLR 2026_

### Official Review · Reviewer_dtZN · 2025-10-29

**Soundness:** 2
**Presentation:** 2
**Contribution:** 2
**Rating:** 4
**Confidence:** 3

**Summary:**

The authors employ a multi-agent framework for hard query generation and use this for contrastive learning for retrieval tasks. The multi-agent framework consists of two steps - a consenus between a code agent and a CoT agent step , a multi-agent group discussion phase (if the first did not lead to a consensus). Theire results show their model improves upon other approaches.

**Strengths:**

1. The paper tackles an important problem.
2. Their agentic solution seems intuitive and makes sense for applications.
3. The need for hard queries and the challenges in building them is present both in research and industry and particularly important for domain heavy topics like medicine, science, engineering, telecom etc.
4. their results are numerically superior.

**Weaknesses:**

The authors measure the performance of their model by precision,recall, NDCG. But the core aspect to be measured is are the queries harder ? Where is this being measured? For example in Table 1, the original data has a lower precision, recall than the model - how do we know that the questions are harder? One possible option is measure on generated queries on pre-trained models and compare with provided data - harder queries should lead to poorer performance.

Overall the paper is interesting but it is hard to understand if the benefits are actually significant or statistical noise and how much of it have the queries improved (become harder)?

Specific comments below

1. What models are used in the different agents. How does the agent choice (size, model family) affect the results has not been analyzed.
2. Numbers in tables need statisical significance testing - is the ContrastGen model statistically better than the second best model? In table 1 for example most improvements are less than 1 percent - this needs statisical validation.
3. Figure 3 exagerrates the differences and no statisical testing is done. For example on top row left (recall@10) all numbers are 52.xx% but the image tends to give an impression of much higher variation. Without statistically testing how many are equal. Authors can report test results from say ANOVA or pairwise t-tests.
4. In table 4, what does the 0 label mean? Do the datasets map "6 quart crackpot" to the text? More likely this is drawn from a negative. if so, is it a negative with high cosine similarity. Without clarity on these aspects it is hard to understand how to interpret this especially the label 0 scenario.
5. Results from ablation study also inform little without statisical testing.
6. The problem of generating hard queries is an important one both in research and in practice (industry). However, what is not clear how this can be extended to domain heavy scenarios. Also are the costs of generating a query of 102k tokens on average prohibhitive for large dataset generations?
7. The datasets are public datasets and hence the LLMs would have seen them. An analysis on propietary data would be interesting here if possible (or future work)


Minor comments:-

1. Lines 91-104 are largely repetitive of content from Section 1.
2. What is $\mathbf{w}$ in $E(.;\mathbf{w})$ is not very clear. Embedding models do not generally have any parameters other than dimension?
3. Typo in table 1 - should be BGE-M4 not BEG-M3.

**Questions:**

Please respond to the weaknesses above.

---

> ### Author Response · Authors · 2025-11-30
>
> Thanks you for your good suggestions. we measure generated queries on pre-trained models and compare it with provided data:  as shown in this table - harder queries lead to poorer performance.
>
> | data type | dataset name  | recall@10 |precision@10 |NDCG@10 |
> | :--- | :--: | ---: | ---: |---: |
> | original data | Shopping Query | 0.5502 | 0.4667 | 0.6567 |
> | hard data | Shopping Query | 0.3988 | 0.0417 | 0.2286 |
> | original data | arxiv | 0.6000 | 0.0600 | 0.3735 |
> | hard data | arxiv | 0.5286 | 0.0529 | 0.2921 |
>
> Weakness 1. We use gpt-4o-mini as our default base language models of agents, what's more. We have tested ContrastGen on three open-source models: llama3.2 3B, llama3.1 8B, llama3.3 70B on shopping queries dataset. llama3.2 3B model has instruction following; the code agent's code writing contains syntax errors, which will reulst in crash of ContrastGen. llama3.3 70B has an advantage in recall compared to llama3.1 8B, but is slightly worse in NDCG. Therefore, ContrastGen requires models with good instruction following, coding writing, and multi role chatting, otherwise, the performance will drop rapidly.
>
> | Embedding model | query gen | discussion | recall@10 |precision@10 |NDCG@10 |
> | :--- | :--: | ---: | ---: | ---: |---: |
> | BGE-M3 | llama3.2 3B | gpt-4o-mini | 0.5303 | 0.3792 | 0.5995 |
> | BGE-M3 | llama3.1 8B | gpt-4o-mini | 0.5338 | 0.3844 | 0.6048 |
> | BGE-M3 | llama3.3 70B | gpt-4o-mini | 0.5349 | 0.3831 | 0.6010 |
>
> Weakness 2. Thank you for your good advice, it's not very common to do statisical significance testing in the field of dense passage retrieval in natural language processing. RocketQA use Recall@k, MRR@k to measure the performance, M3-Embedding use nDCG@K,  Recall@K to measure the performance. we follow the previous work that use the commonly used metric to evaluate the effectiveness of our methods. Due to the fluctuate of every metric of different dataset, so we report the average metric of each method, which can be a way to prove the significance of our methods:
>
> Table 1, row 2 in our paper
> | Method | avg recall@10 | avg precision@10 |avg NDCG@10 |
> | :--- | :--: | ---: | ---: |
> | BM25  | 0.5091 | 0.1433 | 0.4233 |
> | Zero Shot| 0.5392 | 0.1469 | 0.4394  |
> | original data | 0.5342 | 0.1472 |0.44  |
> | CoT | 0.5333 | 0.1462 | 0.4229 |
> | In-Batch Neg | 0.5515  | 0.1546  | 0.4583 |
> | ContrastGen | 0.5650 | 0.1599 | 0.4661 |
>
> Table 1, row 3 in our paper
>
> | Method | avg recall@10 | avg precision@10 |avg NDCG@10 |
> | :--- | :--: | ---: | ---: |
> | BM25  | 0.5091 | 0.1433 | 0.4233 |
> | Zero Shot| 0.6845 | 0.1735 | 0.5285  |
> | original data | 0.6914 | 0.1839 | 0.5615  |
> | CoT | 0.6889 | 0.1813 | 0.5403 |
> | In-Batch Neg | 0.6887  | 0.1848  | 0.5359 |
> | ContrastGen | 0.7022 | 0.1924 | 0.5747 |
>
> ContrastGen can archive more than 1% on avg Recall@10 than the second best model.
>
> Weakness 3. That's a very insightful critique regarding the presentation of results and the need for statistical rigor in scientific papers. You're absolutely right that relying solely on visualized, slightly varying percentages without statistical testing can be misleading.
>
> Since we can't mine enough hard examples from the candidate passages, the candidate passage is usually very distinguishable, the model is struggle to find a hard query, the number of hard sample only takes a very small fraction of the whole dataset. the data used in the figure 3 are mostly easy data, the results shows that solely adding more data can't always increase the performance of a model. From other perspective, this can be a way to verify that more easy data can't improve the performance further.
>
> Weakness 4. The query is "6 quart crockpot", and the passage is "Mellanni Queen Sheet Set - Hotel Luxury 1800 Bedding. Extra soft cooling sheets & pillowcases; deep pocket (up to 16 inch mattress). Wrinkle, fade, stain resistant (4 Piece, Queen, White)." Label 0 indicates that the query and passage are irrelevant. Thank you for reminding us. This is an example of original data that does not have high cosine similarity and is used to mine more negative examples. We will add more description about such cases in the revised version of this paper.
>
> Weakness 5. Thank you for pointing out the statistical perspective of validating the effectiveness of a method, but it is not common in the field of dense passage retrieval in natural language processing, you can refer to the papers, like, M3-Embedding, DPR, RocketQAv2. A common approach is to add more case studies to prove the rationality of our methods. Therefore, we will upload several case studies to help you better understand the basic principle of ContrastGen in the revised version.
>
> DPR. https://aclanthology.org/2020.emnlp-main.550/
> RocketQAv2. https://aclanthology.org/2021.emnlp-main.224/
> M3-Embedding. https://aclanthology.org/2024.findings-acl.137/

---

> ### Author Response · Authors · 2025-11-30
>
> Weakness 6. Thank you for pointing out the application of this method to specific scenarios. Due to the constraint of high costs, we only tested its effectiveness on the embedding retrieval task. However, this approach is well-suited for mining higher-quality data, particularly in scenarios where data is scarce. LLMs with these mechanisms can serve as an alternative if you aim to further improve performance.
>
> Weakness 7. It is interesting that when an LLM performs well on public datasets, we often cannot distinguish between two cognitive processes: generation and memorization. This issue is common across all LLM-based tasks, not merely those involving LLM-based data generation strategies. Although proprietary data is accessible to us, the results are not suitable for reporting in this paper due to a confidentiality agreement.
>
>
> Comments 1: Thanks you, we will update this section in the final revision version.
> Comments 2: currently, the pretrained embedding model are BERT or GPT based structure, which encode the sentence information into a vector,
> the W represents the learnable weights of the pretrained embedding model.
> Comments 3: Thank you for pointing out my typo errors, it's fixed.

---

### Official Review · Reviewer_HeEt · 2025-11-01

**Soundness:** 2
**Presentation:** 2
**Contribution:** 3
**Rating:** 4
**Confidence:** 3

**Summary:**

The paper ContrastGen: A Multi-Agent Contrastive Framework for Hard Retrieval Data Generation introduces a multi-agent approach to generate challenging query–passage pairs for improving embedding-based retrieval models. The framework has two stages: first, a query generation agent rewrites queries while two contrasting evaluators, the Code Agent using rule-based logic and the CoT Agent using semantic reasoning, assess their match with the passage. When the two agents disagree, the sample is identified as a hard example. These hard cases are then refined through a multi-agent group discussion, where several expert agents debate and vote to decide the final label. This process produces training data that captures nuanced reasoning and complex semantics, enhancing the model’s ability to distinguish relevant from irrelevant passages. Experiments on Shopping Queries, arXiv, and MS MARCO datasets show consistent improvements in retrieval metrics, and ablation studies confirm that both the dual-agent contrast and the discussion mechanism are essential. The main contribution is reframing retrieval data generation as a multi-agent contrastive process that uses agent disagreement and collaboration to create high-quality hard samples that improve model robustness.

**Strengths:**

1) Instead of following the conventional path of improving model architectures, the authors take a data-centric approach by generating high-quality hard samples to enhance the discriminative capability of retrieval models.

2) The idea of defining sample difficulty through the disagreement between a rule-based Code Agent and a reasoning-based CoT Agent is both intuitive and interesting.

3) The authors validate the framework across multiple public datasets and diverse embedding models, consistently showing performance gains.

**Weaknesses:**

1) The system’s performance is tightly coupled to specific prompts, role descriptions, and reasoning styles. The lack of a systematic sensitivity study leaves open whether the method would remain stable under different model versions, prompting templates, or domain shifts. Besides, the final label aggregation relies on an ad-hoc greedy rule that mixes majority votes with the first CoT decision. This design lacks theoretical grounding and could bias results toward early, possibly noisy agent judgments.

2) The observed performance peak at a moderate amount of generated data is not theoretically or empirically explained. A deeper analysis of data diversity, redundancy, or quality filtering would clarify why excessive generation degrades results.

3) The reported improvements lack confidence intervals or significance tests, and the authors do not provide the code, making it difficult to assess robustness or reproducibility.

**Questions:**

Q1: How does the proposed pipeline scale when generating millions of samples?

Q2: What is the average time and cost per generated example, and how does that compare to human annotation or standard augmentation methods?

Q3: Did you test other aggregation mechanisms, such as weighted confidence or probabilistic fusion? And how do you handle ties or inconsistent reasoning paths across discussion rounds?

Q4: What are the most common failure modes for the Code Agent and CoT Agent?

Q5: Did you observe any systematic biases in the group discussion outcomes (e.g., majority bias or echoing effects)?

---

> ### Author Response · Authors · 2025-11-28
>
> Weakness 1. ContrastGen address the ability of code-based reasoning, text-based complex task reasoning, group chat-based disscussion. For any LLM-based task, it's obvious that a good prompt, suitable role descriptions, and appropriate reasoning styles are crucial for performance improvement, unless the LLMs themselves are super powerful on human language understanding.
>
> as shown in the table below, we have tested the results from three open-source models (diferent model versions): llama3.2 3B, llama3.1 8B, llama3.3 70B. llama3.2 3B model has poor instruction following ability; the Code Agent’s code writing contains many syntax errors, which cause ContrastGen to crash. llama3.3 70B has an advantage in recall compared to llama3.1 8B, but is slightly worse in NDCG. Therefore, ContrastGen requires models with good instruction following, coding writing, and multi role chatting, otherwise, the performance will drop rapidly. what's more, ContrastGen performs well on gpt-4o-mini, because gpt performs good in instruction following and coding. In total, ContrastGen performs well on LLMs with relative good code writing and instruction following ability.
>
> | PELM | query gen | discussion | recall@10 |precision@10 |NDCG@10 |
> | :--- | :--: | ---: | ---: | ---: |---: |
> | BGE-M3 | Llama3.2 3B | gpt-4o-mini | 0.5303 | 0.3792 | 0.5995 |
> | BGE-M3 | Llama3.1 8B | gpt-4o-mini | 0.5338 | 0.3844 | 0.6048 |
> | BGE-M3 | Llama3.3 70B | gpt-4o-mini | 0.5349 | 0.3831 | 0.6010 |
> | BGE-M3 | GPT-4o-mini | GPT-4o-mini |0.5565 | 0.4104 | 0.6362 |
>
> As mentioned in our paper, we have tested our methods on three scenarios (the robustness and generation of ContrastGen): shopping goods search, arXiv papers search, and general web search. Each scenario has its corresponding prompt template, and a lightweight modification of the template is necessary. For example, in shopping scenarios, the passage includes titles, bullet points, SKU information, and so on; in arXiv papers scenarios, the passage contains titles, authors, abstracts, publication years, etc. Using a unified prompt template is not practical due to the significant differences between the domain datasets.
>
> Weakness 2. Thank you for pointing out this important issue. However, it's not a special case in embedding retrieval training, but rather an open problem: the performance of a model doesn't always monotonically increase as expected when adding more training data. The degradation in results when using excessive data, especially generated data, is likely due to the introduction of low-quality, redundant, or misleading samples. These samples ultimately confuse or bias the model, outweighing the benefits of a larger sample size. Since Agents have a strong dependency on the inherit ability of the base large language model. We assume that this phenomenon could be alleviated by utilizing a superior base model, such as Gemini 3 Pro, GPT-5, or others. However, verifying this empirically would be infeasible due to the high cost (requiring thousands of US dollars). Therefore, how to generate high quality data with a weak dependency on the base large language model is interesting, we will reserve this for future work.
>
> Weakness 3. One way for reproducibility of ContrastGen it to set fixed random seed, and we implement our embedding training methods based on the public HuggingFace's transformer library, we set fixed seed in our source code:
>
> ```
> random.seed(42)
> np.random.seed(42)
> torch.manual_seed(42)
> torch.cuda.manual_seed(42)
> ```
> In general, our methods will be influenced by the quantity, the ratios of the positives and negatives, the ratios of hard data and easy data, all the possible factors on the performance are reported on the paper. the roubustness can be drawn from the other perspectives, for examples, we tested our methods on three different scenarios, e-commerce, academic field and general web search engine, which can be an alternative way to verify the robustness of our methods.
>
> Finally, we will release our source code under an open-source license (e.g., MIT License), and make it public available if this paper is accepted.

---

> ### Author Response · Authors · 2025-11-28
>
> Q1: ContrastGen addresses capabilities in code-based reasoning, text-based complex task reasoning, and group chat-based discussion. One way to apply this mechanism to any data size is to use open-source models as a replacement. However, closed-source models generally perform better than open-source ones in general. For example, Llama-based models initially failed to directly support multi-round conversations involving different roles, largely because their supervised fine-tuning (SFT) did not include multi-person, multi-round dialogue data. This limitation makes it challenging to replace closed-source models with open-source alternatives. Furthermore, we plan to adapt this strategy to most mainstream open-source models in the future. What's more, it is fit for scenarios that the data is scarse, and the embedding retrieval performance is not good, applying Contrastgen to generate and mining more high quality is feasible.
>
> Q2: The average time for generating a hard example is 30 seconds, while the average time for generating an easy example is 12.5 seconds. Due to the sparsity and scarcity of hard data, easy samples are also added to the generated dataset with a certain probability to increase its diversity. The average cost for generating an easy example is 0.0023 US dollars, and the average cost for generating a hard example is 0.0227 US dollars.
>
> It is not feasible to compare with human annotation because it is labor-intensive and time-consuming. One hypothesis is that we can achieve equal or even better performance than human annotation, since humans often make mistakes or produce contradictory data due to differences in knowledge and experience. Furthermore, we have tested two standard augmentation methods, as shown in Table 1 of our paper. The typical standard data augmentation method in information retrieval is in-batch negatives training (referred to as "In-Batch Neg" in our paper), which dynamically selects negative samples—a form of data augmentation—from the same training batch to enhance discriminative capability. Moreover, another method is LLM-based data augmentation (referred to as "COT" in our paper), where we use LLMs to generate multiple queries for a passage and instruct the LLMs to determine whether the query and passage match.
>
> Q3: It's very interesting to use multiple aggregation mechanisms, our greedy vote strategy is a way of weighted vote, because we implicitly add more weight to the cot's decision during the vote decision process. what's more, since agents can sometimes be overconfident in their results, how can we obtain the reliable confidence or probability score for each agent? overconfidence may introduce bias, leading to inaccurate confidence or probability scores.
>
> Within each group, we use a majority vote to reach a final decision. To prevent a tie during discussions, each group consists of an odd number of agents. Every agent expresses its reasoning and preliminary decision. Subsequently, other agents will either agree or disagree with the previous decisions, clarifying their own reasons and final votes.
> Extremely, if two of them insist on their decision, no answer will output from this group, there are more than 1 discussion groups, if multi groups reach a tie, no output is given, the CoT Agent's decision will be adopted as the final label.
>
> Q4: ContrastGen has a dependency on the coding and reasoning ability of a large language model, if a large language model performs poor in instruction following and coding, the contrast mechainsms between Code Agent and Code Agent may be useless. So we recommend using models that are exel at reasoning the coding to applying this mechanism.
>
> Q5: Yes, Group discussion involves multiple agents engaged in a conversation. This type of interaction is not inherently designed for collective reasoning; rather, it resembles a social behavior that may follow freestyle. For example, if one agent is influential and insists on its opinion, the other agents may follow that opinion without doing further thinking or analysis, which can affect the final decision. probabily, Using a majority vote may further amplify such errors.

---

### Official Review · Reviewer_71BY · 2025-11-02

**Soundness:** 2
**Presentation:** 2
**Contribution:** 2
**Rating:** 4
**Confidence:** 4

**Summary:**

While I appreciate the clear problem formulation and insightful process-level metrics, I believe the paper requires several revisions. My primary concerns are the questionable novelty of the pipeline (Sec.3) and the contradiction between the claim of a "long" benchmark and its actual length. Additionally, crucial methodological details regarding data filtering, utility calculation, and the justification for your similarity metrics are insufficiently explained.

**Strengths:**

Clear Problem Formulation. The authors clearly identify a critical gap in existing research: the lack of benchmarks for evaluating long-chain, structured, and agentic multimodal reasoning. Current benchmarks are often limited to 1-2 hop retrievals, which is insufficient for testing advanced agentic capabilities.

Comprehensive Evaluation and Metrics: The paper moves beyond simple final-answer accuracy by introducing insightful process-level metrics like Hit per Step (HPS) and Rollout Deviation (RD).

**Weaknesses:**

1. Novelty of Sec.3.1. Could the authors elaborate on the novelty of "Agentic MM-RAG Pipeline" compared to [OmniSearch (Sec.4.2)](https://arxiv.org/pdf/2411.02937)?
2. The authors claim to introduce the "first benchmark with **long**, step-wise annotated instructions.(Line101)" However, the provided average length of 3.7 steps seems to contradict the descriptor "long."
3. Line180, Equation 2. The calculation of $Util(t)$ will become expensive if $T$ is large or the base model is large. (Did I miss some clever way to efficiently calculate $Util(t)$?)
4. Line197: The logic of marking a sample as *redundant* seems not rigorous enough. E.g., if a redundant and useless entity appears twice in a trajectory, which is likely when models make a mistake, the data will still be marked *not redundant* because $Nav(t)$ equals 1.
5. Need more elaboration on Data Filtering (Line175). What happens if a step is marked as redundant? Is it removed directly? How do you handle the connection of context of this removed step? Is direct concatenation likely to cause incoherent logic?

**Questions:**

1. Eq.4 How do you define the equality of $\hat{r}_{t'}$ and $r_{t}$? (E.g., exact match?) The evidence could be sequences of tokens.

2. Line233 The paper describes each reasoning graph as "a sequence of retrieval-augmented reasoning steps indexed by t." This phrasing suggests a strictly linear, sequential structure, essentially a reasoning chain. However, the term graph typically implies the possibility of branching, merging, or more general topological structures beyond a simple sequence.
    - Could the authors clarify whether the "reasoning graphs" in this work are always linear chains (i.e., sequences), or whether there are scenarios in which they exhibit genuine graph-like structures (e.g., multiple predecessors/successors for a step, parallel reasoning paths, or dynamic step dependencies)?
    - If all reasoning graphs are indeed sequences, would it be more precise to refer to them as reasoning chains to avoid potential confusion? Conversely, if non-linear graph structures do arise, please elaborate on (1) under what conditions such structures occur, and (2) how the model's inference procedure generates them.

3. Could you please elaborate on why applying *maximum-weight bipartite matching* is a good enough way to model the similarity of two trajectory graph.

---

> ### Author Response · Authors · 2025-11-27
>
> Weakness 1. Our primary objective is to improve the quality of retrieval data generated by Large Language Models (LLMs). We acknowledge that relying solely on LLMs to evaluate the relevance between a query and a document is not always reliable, and LLMs often struggle to process complex data. Consequently, we introduce a broad definition of 'hard samples,' and propose a contrast mechanism to help identify these complex instances. If these hard samples (where LLMs are prone to errors) are properly identified and corrected, data quality can be significantly enhanced.
>
> Weakness 2. As shown in the table below, when the number of groups is 2 with the CoT Agent's result, the final decision process is theoretically equivalent to a majority vote among three groups. what's more, performance improves when we treat the chain-of-thought (CoT) as an additional group in the majority vote. Since expanding the number of groups to at least four is impractical—as it would cost thousands of US dollars to apply these mechanisms to all the data mentioned in this paper—treating CoT as a group for majority voting can significantly reduce token consumption and improve performance. Another issue with majority voting arises when there are four groups: if two groups agree and two disagree, it becomes impossible to reach a final decision.
>
> | PELM | LLM name | group num | with cot | recall@10 |precision@10 |NDCG@10 |
> | :--- | :--: | ---: |---: | ---: |---: |---: |
> | BGE-M3 | GPT-4o-mini | 3 | no |0.5336 | 0.3792 | 0.5986 |
> | BGE-M3 | GPT-4o-mini | 2 | yes |0.5391 | 0.3857 | 0.6089 |
>
> Weakness 3. ContrastGen addresses capabilities in code-based reasoning, text-based complex task reasoning, and group chat-based discussion. One way to apply this paradigm to any data size is to use open-source models as a replacement. However, as is widely recognized, closed-source models generally perform better than open-source ones in general. For example, Llama-based models initially failed to directly support multi-round conversations involving different roles, largely because their supervised fine-tuning (SFT) did not include multi-person, multi-round dialogue data. This limitation makes it challenging to replace closed-source models with open-source alternatives. Furthermore, we plan to adapt this strategy to most mainstream open-source models in the future.
>
> Weakness 4. We simply append role information, chat history, and the next speaker’s task into a single prompt. here are the prompt of selector:
>
> ```
> You are in a role play game. The following roles are available:
> {{roles}}.
> Read the following conversation.
> Then select the next role from {{agentlist}} to play. Only return the role.
> ```
> then we use the validator to guide the speaker selection process, if the selected speaker is valid, then it will end the speaker selection process, if not, the validator
> will remind the selector to select a valid agent name.
> is:
>
> ```
> You didn't choose a speaker. As a reminder, to determine the speaker use these prioritised rules:
>  1. If the context refers to themselves as a speaker e.g. "As the..." , choose that speaker's name
>  2. If it refers to the "next" speaker name, choose that name
>  3. Otherwise, choose the first provided speaker's name in the context
>  The names are case-sensitive and should not be abbreviated or changed.
>  The only names that are accepted are {agentlist}.
>  Respond with ONLY the name of the speaker and DO NOT provide a reason.
> ```
> more details can be found in Dynamic Group Chat section of the paper [AutoGen: Enabling Next-Gen LLM Applications via Multi-Agent Conversation](https://arxiv.org/abs/2308.08155)
>
> Q1: We will upload a case that shows an inconsistency between the code agent and the chain-of-thought (CoT) agent. The code agent tends to write Python code to verify the relevance between the query and the passage, which reflects its rule-based ability—being skilled at calculating and enumerating all possible solutions. On the other hand, the CoT agent uses step-by-step reasoning, which is skilled at semantic reasoning, to arrive at the final result, representing the complex analytical ability of a human.
>
> However, the results from large language models are not always reliable due to the inevitable hallucinations that occur. The code agent is less susceptible to hallucinations because it obtains the final answer by executing Python code—as long as the code is correct. In contrast, the CoT agent is more easily affected by hallucinations since it relies on text-based reasoning to reach the final answer.
>
> Q2: It would be interesting to incorporate more agents for relevance judgment, such as a search agent. However, the search results are not always useful for judging the match between the query and the passage, as they may introduce additional noise and increase the difficulty of relevance judgment.
>
> Q3: Thank you for pointing out our typo errors, we have correctd in the revisited edition.

---

### Author Response · Authors · 2025-12-02

Since all the reviews for ICLR 2026 were reverted on November 28, and Reviewer 71BY mixed up their review with another paper before November 11, we have updated Reviewer 71BY's response for your reevaluation of this paper.

Summary:

This paper is well-motivated, presenting a novel perspective for improving retrievers by tackling hard tasks. We are convinced by the sufficient baselines used to verify the proposed method and believe it inspires future work on more challenging tasks. I have several minor concerns, including potential unreliability in using LLMs for relevance evaluation, the lack of a simpler majority vote baseline, prohibitive computational costs, and some lack of clarity in the agent selection process. We further seek clarification on case analyses for hard samples, the potential benefit of incorporating more specialized agents.

Soundness: 3: good

Presentation: 3: good

Contribution: 3: good

Strengths:

This paper is well-motivated. It tries to solve hard tasks in a new perspective of view, improving retriever. Sufficient baseline retrievers adopted to verify the proposed method to improve the retriever. This work inspires the future direction on solving more challenging tasks like BrowseComp.

Weaknesses:

1. Previous work has shown that asking LLMs themselves to evaluate the relevance of queries and documents are not that reliable [1](https://arxiv.org/abs/2505.21870). Applying Code Agent and CoT Agent is also within this paradigm. It would be better if there are experiments conducted to verify the relevance between inconsistency (among Code and CoT Agent) and query difficulty.

2. The initial CoT agent's decision $y_{0}^g$. is only overturned if all L discussion groups unanimously disagree. Thus, it seems that a simpler majority vote is a more standard and intuitive baseline, but no comparison is offered.

3.  Prohibitive computational cost makes the approach infeasible for generating datasets of any significant size. (1 hard example=~58 API calls and 100k tokens)

4. Lack of Clarity on Agent Selector and Validator.

Questions:

1. Line186 & 208. Inconsistency among Code Agent and CoT Agent indicates a hard sample. Could the author give any case analysis?

2. Related to Weakness1, will it be more trustworthy by incorporating more Agents besides Code Agent and CoT Agent? E.g., agents that specialize searching. Or, the current setup is enough.

3. Typo? Line243: $y^t_{i,k}$ and $y^t_{i,k}$

Flag For Ethics Review: No ethics review needed.

Rating: 6: marginally above the acceptance threshold. But would not mind if paper is rejected

Confidence: 4: You are confident in your assessment, but not absolutely certain. It is unlikely, but not impossible, that you did not understand some parts of the submission or that you are unfamiliar with some pieces of related work.

Code Of Conduct: Yes

---

### Meta-Review · Area_Chair_jf4U · 2026-01-09

**Summary:**

This paper proposes a multi-agent framework to generate hard queries for improving the performance of the embedding models. This manuscript holds the view that generating queries that retain semantics and logical knowledge similar to those of human-generated queries could help to enhance the training performance of the embedding model. The key idae of this study lyes on the data-level persepctive on tackling the performance bottleneck and effectiveness limitation. The strengths and the weaknesses are listed comprehensively in the reviews so I will not further point them out.
Basically, the reviewers hold a relevantly consist view that this paper might be marginally around the acceptance threshold (3 below, 1 above). After carefully reading the manuscript and the reviews, I agree with the reviewer's concerns on the statisical testing,  the correlation between the hard examples and data quality, the voting strategies, computation cost, and so on. And it seems  that the mentioned concerns are not well addressed.

**Reviewer Concerns:**

Since the discussion of this paper is not sufficient, I leave it blank.

**Reviewer Scores:**

I think most of the reviews from Reviewer 71BY are not addressed (weaknesses 1, 2, 3). Some of the reviews from the others are not well addressed.

---

### Decision · Program_Chairs · 2026-01-26

Reject